# Crosstalk Between Omental Adipose-Derived Stem Cells and Gastric Cancer Cells Regulates Cancer Stemness and Chemotherapy Resistance

**DOI:** 10.3390/cancers16244275

**Published:** 2024-12-23

**Authors:** Jun Kinoshita, Kenta Doden, Yusuke Sakimura, Saki Hayashi, Hiroto Saito, Toshikatsu Tsuji, Daisuke Yamamoto, Hideki Moriyama, Toshinari Minamoto, Noriyuki Inaki

**Affiliations:** 1Department of Gastrointestinal Surgery, Kanazawa University, Kanazawa 920-8641, Japan; k.doden@med.kanazawa-u.ac.jp (K.D.); yusukesakimura@staff.kanazawa-u.ac.jp (Y.S.); saki.n0806@gmail.com (S.H.); masaeritomo16@staff.kanazawa-u.ac.jp (H.S.); tsuji104@staff.kanazawa-u.ac.jp (T.T.); yamamoto-daisuke@staff.kanazawa-u.ac.jp (D.Y.); hidemori@med.kanazawa-u.ac.jp (H.M.); n.inaki@med.kanazawa-u.ac.jp (N.I.); 2Japan Community Health Care Organization Kanazawa Hospital, Kanazawa 920-8610, Japan; minamoto@staff.kanazawa-u.ac.jp; 3Department of Molecular and Cellular Pathology, Kanazawa University, Kanazawa 920-8640, Japan

**Keywords:** gastric cancer, adipose-derived stem cell, peritoneal metastasis, cancer-associated fibroblast, cancer stem cell

## Abstract

Peritoneal metastasis, where cancer spreads to the lining of the abdomen, is a serious issue for patients with gastric cancer and often affects adipose-rich areas like the omentum. This study explores whether stem cells from the omentum contribute to peritoneal metastasis by becoming members of the cancer microenvironment and affecting cancer stem cells. By examining how these stem cells interact with gastric cancer cells, this study finds that they can promote peritoneal metastasis and resistance to treatment. These findings lead to new strategies for targeting omentum-derived stem cells to improve treatment outcomes for patients with peritoneal metastasis.

## 1. Introduction

Peritoneal metastasis (PM) is a common form of metastasis and recurrence in patients with gastric cancer (GC). Despite various approaches to treatment being attempted, such as intraperitoneal chemotherapy (IPEC), hyperthermic IPEC (HIPEC), and immunotherapy, PM remains a major challenge in patients with GC [1]. Therefore, elucidating the molecular and cellular mechanisms underlying tumor progression in PM is necessary for developing novel strategies for the prevention and treatment of this devastating condition.

Resistance to anoikis is a hallmark of PM, allowing free cancer cells that have migrated from the primary tumor to the peritoneal cavity to survive in an anchorage-independent manner [2]. These cells must create a unique tumor microenvironment (TME) susceptible to metastasis known as a premetastatic niche [3]. PM preferentially occurs in adipose-rich organs such as the omentum. Premetastatic niches created by intraabdominal adipose tissue may potentially create fertile microenvironments for progression to PM. To verify the mechanism by which adipose tissue is susceptible to PM, we previously revealed that intraabdominal hypoxic conditions enhance CD36 expression in GC cells and promote migratory and invasive abilities through CD36-mediated intracellular uptake of the free fatty acids secreted by adipocytes [4]. In addition to adipocytes, adipose-derived stem cells (ASCs), isolated from the stromal vascular fraction of adipose tissue, are attracting attention in regenerative medicine because of their ability to differentiate into multilineage cells. However, conflicting evidence has emerged in recent years regarding the safety profile of ASC applications, with a potential pro-oncogenic effect of ASCs on various cancer types having been demonstrated [5,6]. ASCs have been suggested to differentiate into tumor-promoting stromal cells which are known as cancer-associated fibroblasts (CAFs) [7]. Multiple lines of evidence indicate that CAFs can remodel the extracellular matrix structure and TME, contributing to the generation of the cancer stem-like cell (CSCs) niches through paracrine factors [8,9]. Additionally, several reports suggest that ASCs differentiated into CAFs in the TME play a critical role in chemotherapy resistance and relapse [10,11].

These observations suggest that ASCs derived from intraabdominal adipose tissue may contribute to the development of the CSC niche and the treatment resistance of PM against GC; however, the underlying mechanisms have not been elucidated. This study aimed to investigate whether ASCs isolated from the human omentum can act as progenitors of CAFs when co-cultured with GC cells. We also analyzed the effects of ASCs on the cancer stemness and chemoresistance of GC cells in vitro and in vivo.

## 2. Materials and Methods

### 2.1. Cell Lines and Cell Culture

The human GC cell lines, MKN45, MKN74, and NUGC4, used in this study were purchased from the Japanese Collection of Research Bioresources Cell Bank (Osaka, Japan). Cells were maintained in an RPMI-1640 medium supplemented with 10% fetal bovine serum (FBS).

### 2.2. Isolation and Expansion of Omental ASCs

Grossly normal human omental tissues were obtained from five patients who underwent gastrectomy for GC. None of the patients had peritoneal metastases at the time of laparotomy. All patients provided written informed consent, and the study was conducted according to institutional guidelines under an approved protocol. ASCs were isolated from omental tissues, as described previously [12,13]. Briefly, freshly collected omental tissues were washed with sterile phosphate-buffered saline (PBS) seven times and minced into small pieces. Tissue pieces were incubated with 1 mg/mL collagenase type I (Sigma-Aldrich, St. Louis, MO, USA) in DMEM/F12 (Gibco, Waltham, MA, USA) for 1 h at 37 °C. After centrifugation at 300 g for 10 min, the cellular pellet was resuspended in PBS and filtered through a 100-μm mesh filter to remove undigested adipose tissue. The filtrate was centrifuged again, and the resulting pellet was seeded onto 6 cm cell cultures plates in a culture medium (DMEM/F12 containing 10% FBS). ASCs isolated from omental adipose tissue were expanded in expansion medium containing 5 ng/mL of epidermal growth factor (EGF), 1 ng/mL fibroblast growth factor (FGF), and 0.25 ng/mL of transforming growth factor-β1 following the established protocol [12,13]. Early passages (P2–P5) of isolated ASCs were used in this study.

### 2.3. Flow Cytometry

For cell surface marker analysis, ASCs from the second passage were harvested by trypsinization in blocking buffer solution containing 1.5% bovine serum albumin (BSA). Cells were incubated with fluorescein-conjugated antibodies (CD44 (11-0441-82, eBioscience, San Diego, CA, USA), CD90 (11-0909-71, eBioscience), CD45 (560915, BD pharmingen, San Diego, CA, USA), or Epcam (324210, BioLegend, San Diego, CA, USA)) or stained with a phycoerythrin-conjugated CD105 (12-1057-41, eBioscience), CD14 (325603, BioLegend), CD34 (12-0349-41, eBioscience), or CD29 (11-0299-71, eBioscience), as recommended by the manufacturer. The appropriate fluorescein- or phycoerythrin-conjugated isotypes were used as controls in each assay. The cells were analyzed by flow cytometry using an Attune acoustic cytometer (Applied Biosystems, Waltham, MA, USA; Life Technologies, Carlsbad, CA, USA). Data were transferred and reanalyzed using the FlowJo software V9 (Tree Star, Ashland, OR, USA).

### 2.4. Cell Co-Culture Studies

ASCs and GC cells were indirectly co-cultured using transwell inserts equipped with a 0.4-μm pore size membrane (Corning, Corning, NY, USA). In brief, 5 × 10⁵ ASCs were placed in the upper chamber and 1 × 10⁵ cancer cells were seeded in the lower chamber. After 3 days of co-culturing, the cancer cells were collected for analysis. This indirect co-culture method was utilized for both the multiplex cytokine/chemokine arrays and Western blotting assays. For the IL-6 neutralization assay, ASCs and GC cells were co-cultured indirectly in transwell inserts, with 0.1 μg/mL of human IL-6 neutralizing antibody (InvivoGen, San Diego, CA, USA) being added to the culture.

### 2.5. Human Chemokine Array

The co-culturing of ASCs and GC cell lines (MKN45, MKN74, and NUGC4) was performed in a 6-well transwell assay. The conditioned medium from ASCs (2 mL) was harvested after 7 days of culturing. The levels of cytokines, growth factors, and chemokines in the conditioned medium were measured using the Proteome Profiler Human XL Cytokine Array Kit (ARY005B, R&D Systems, Santa Clara, CA, USA) following the manufacturer’s protocol. In brief, the array membranes, which were pre-spotted with capture antibodies for specific target proteins, were incubated overnight at 4 °C with conditioned media that had been pre-treated with a biotinylated detection antibody cocktail. After incubation, the membranes were washed and treated with streptavidin–HRP for 30 min at room temperature. Following further washing steps, the proteins bound to the membranes were detected using chemiluminescence reagents. The chemiluminescent signals were captured using the Gel Image system (ATTO, Light-Capture II, Chuo City, Japan) and analyzed with CS analyzer software (Version 3.0) by quantifying the intensities of the detected spots.

### 2.6. Preparation of Conditioned Medium

A conditioned medium (CM) from GC cells was prepared following the method described previously [14]. In summary, 1.0 × 10⁶ cells were plated in 10 cm tissue culture dishes containing 10 mL of RPMI or DMEM with 10% FBS and incubated at 37 °C for 3 days. Afterward, the cells were rinsed twice with PBS and further incubated with 5 mL of serum-free RPMI or DMEM for 2 days. The CM was collected from each dish, centrifuged at 1000× *g* for 5 min at 4 °C, and the supernatant was immediately used as a CM.

### 2.7. ELISA

To determine IL-6 concentrations in ASC culture media, we employed a commercially available Human IL-6 Quantikine ELISA kit following the protocol provided by the manufacturer (R&D Systems). ASCs were maintained in culture conditions with and without the addition of a GC cell-derived CM. After a 48 h incubation period following CM introduction, we assessed the IL-6 levels. Absorbance measurements were taken at wavelengths of 450 and 570 nm using a Multiskan GO microplate spectrophotometer (Thermo Scientific, Waltham, MA, USA). IL-6 concentrations were then derived from these absorbance readings by referencing a pre-established standard curve.

### 2.8. Sphere Formation Assay

In the study, 1 × 10^3^ ASCs, 1 × 10^3^ GC cells, or 1 × 10^3^ GC cells plus 1 × 10^3^ ASCs were incubated for 10 days in 1 mL of modified sphere medium (DMEM/F12 medium supplemented with 1X B-27 supplement [Thermo Fisher Scientific, USA]), 20 ng/mL EGF (PeproTech, Cranbury, NJ, USA), 10 ng/mL (FGF)-basic (PeproTech USA), and 4 mg/mL insulin (Sigma-Aldrich) in ultra-low-attachment 24-well plates (Corning Life Sciences, USA). Spheres (>50 μm diameter) in ten random fields per each well were counted and photographed.

### 2.9. MTT Assay

To compare the proliferation rates of ASCs and MKN45 cells, both cell types were seeded separately in 96-well plates at a density of 2000 cells/well. Cell viability was measured at 0, 24, 48, and 72 h using the MTT assay (3-(4,5-dimethylthiazol-2-yl)-2,5-diphenyltetrazolium bromide) as described below.

To investigate the effects of ASCs on the proliferation of GC, MKN45, MKN74, and NUGC4 cells were cultured in a co-culture model. GC cells were placed in the bottom chamber of a transwell with a 0.4 μm pore size membrane while ASCs were cultured in the upper chamber. The MTT assay was used to determine cell viability. GC cells were seeded in 96-well plates at a density of 2000 cells/well, with ASCs being added to the upper chamber in equal numbers.

After the incubation period, the medium was discarded and an MTT solution was added to each well (final concentration, 500 μg/mL), followed by incubation at 37 °C for 3 h. Subsequently, the solution was removed, and 150 μL of dimethyl sulfoxide (DMSO; Wako, Japan) was added to each well. The absorbance of the resulting solution was measured at 535 nm using a Bio-Rad 550 microplate reader (Bio-Rad, Hercules, CA, USA). Cell viability was calculated as the ratio of absorbance in experimental wells to that in control wells. The experiment was repeated independently at least three times.

### 2.10. Western Blotting

Cells were harvested and lysed using a RIPA buffer (Wako) composed of 50 mM Tris-HCl (pH 8.0), 150 mM NaCl, 0.5% sodium deoxycholate, 0.1% SDS, and 1.0% NP-40 substitute, supplemented with a 1% protease inhibitor cocktail (Sigma-Aldrich). A BCA protein assay kit (Pierce Biotechnology, Appleton, WI, USA) was employed to determine the protein concentration of each sample. Whole cell lysates were then prepared in a denaturing SDS sample buffer and separated by SDS-polyacrylamide gel electrophoresis using equipment from ATTO Co., Ltd., Japan. For visualization, immunoblots were developed using the ECL Plus kit (GE Healthcare, Buckinghamshire, UK). Among the primary antibodies utilized in this study was an antibody against CD44 variant 6 (CD44v6) (MA54; 1:500; ThermoFisher Scientific, Waltham, MA, USA), Nanog (3580S, diluted 1:5000; Cell Signaling Technology, Danvers, MA, USA), Oct4 (2750S, diluted 1:1000; Cell Signaling Technology), SOX2 (2748S, rabbit, diluted 1:1000; Cell Signaling Technology), GAPDH (016-25523, diluted 1:2500; FUJIFILM Wako Pure Chemical Corporation, Osaka, Japan), and β-actin (4967, diluted 1:1000; Cell Signaling Technology). Expression of β-actin was examined as a loading control. Human recombinant IL-6 was purchased from R&D Systems Inc. and reconstituted in RPMI-1640 medium at 50 ng/mL for its functional analysis.

### 2.11. Immunocytochemistry

To examine the expression of fibroblast activation protein (FAP) α and α smooth muscle actin (αSMA) in ASCs, the cells were cultured on collagen type I-coated four-well slides (BD BioCoat, Horsham, PA, USA). Upon reaching pre-confluence, cells were fixed using an equal parts mixture of methanol and acetone for 10 min. Subsequently, the slides were submerged in methanol with 0.3% H2O2 for 30 min, followed by blocking with 3.3% normal goat serum in PBS. The samples were then incubated overnight at 4 °C with primary antibodies against FAPα (ab53066, 1:100 dilution; Abcam, Cambridge, UK) and αSMA (ab7817, 1:100 dilution; Abcam). To visualize the immunoreactivity, slides were treated with Alexa Fluor 488-conjugated anti-mouse IgG and Alexa Fluor 594-conjugated anti-rabbit IgG secondary antibodies (1:400 dilution; Molecular Probes, Eugene, OR, USA/Invitrogen, Waltham, MA, USA) for 1 h at room temperature. Nuclear staining was performed by incubating the cells with DAPI for 5 min. An immunofluorescence microscope (BX50/BS-FLA; Olympus, Tokyo, Japan) was used to analyze and capture the images.

### 2.12. Histological, Histochemical, Immunohistochemical and Immunofluorescence Examinations

Each subcutaneous tumor specimen removed from xenografted mice was divided into two parts, one of which was shock-frozen in liquid nitrogen and cryosectioned while the other was fixed in 10% neutral buffered formalin and embedded in paraffin. Frozen sections mounted on glass slides were examined by fluorescence microscopy using a standard filter setup to observe PKH26-positive cells. Paraffin sections were stained with hematoxylin and eosin or Mallory-Azan stain and observed under a light microscope. Fibrosis was analyzed by measuring azan (blue)-stained areas on a video display (magnification, ×200) in a blinded manner using a BZ-9000 BZII microscope (Keyence, Osaka, Japan). Two sections were randomly selected from each sample, and three fields from each section were evaluated. The paraffin sections were immunostained with antibodies to αSMA (M085129, diluted 1:100; Dako), IL-6 (ab6672, diluted 1:500; Abcam) overnight at 4 °C. Sections were treated with the EnVision reagent (Dako, Glostrup, Denmark) for visualization. For detection of stemness, deparaffinized sections were incubated with anti-human CD44v6 (33-6700, diluted 1;500; Invitrogen, USA) and anti-Nanog (4903, diluted 1;500; Cell Signaling Technology) in a diluent of PBS with 1% BSA and 0.1% Triton X-100 at 4 °C overnight. Immunostaining was visualized using Alexa Fluor 488 (green)-labeled anti-mouse IgG and Alexa Fluor 594 (red)-labeled anti-rabbit IgG. Nuclei were counterstained with DAPI. To detect apoptosis, a terminal deoxynucleotidyl transferase-mediated dUTP nick-end labeling (TUNEL) immunofluorescence was performed. Deparaffinized sections were treated with the DeadEnd Fluorometric TUNEL system (G3250, Promega, Madison, WI, USA) to detect apoptosis according to the manufacturer’s instructions. Subsequently, sections were stained with propidium iodide solution (1 ug/mL) for 15 min. The images were analyzed using an immunofluorescence microscope (BX50/BS-FLA; Olympus, Tokyo, Japan).

### 2.13. Mouse Xenografts and Chemotherapy

All the mouse experiments were approved by the Institutional Animal Care and Use Committee of Kanazawa University. For subcutaneous flank tumors, we used BALB/c nu/nu mice (female, 4–6 weeks old; Charles River Laboratories Inc., Kanagawa, Japan) as a xenograft model, to investigate the effects of co-culturing MKN45 cells with ASCs. ASCs were pre-stained with the red fluorescent dye PKH26 cell linker kit (Sigma-Aldrich) according to the manufacturer’s instructions. The concentration of PKH26 during incubation was 4 μM. Each of the six control mice was inoculated with 7 × 10^6^ of MKN45 cells in 100 μL of RPMI-1640 subcutaneously, whereas each of the six experimental mice was inoculated with 5 × 10^6^ of MKN45 cells plus 2 × 10^6^ of ASCs in 100 μL of RPMI-1640.

For in vivo chemotherapy experiments, nude mice were subcutaneously inoculated with 7 × 10^6^ MKN45 cells or 5 × 10^6^ MKN45 cells and 2 × 10^6^ ASCs. Mice were assigned to control (six mice per group) and 5-fluorouracil (5-FU) groups (six mice per group). When tumors reached 100–150 mm^3^ in volume, mice received 20 mg/kg 5-FU or carrier (PBS) injected intraperitoneally five times per week for two weeks. Tumors were monitored and recorded every 2 days for 2 weeks, and tumor volume (TV) was calculated using the following formula: TV = length × (width)^2^ × 0.52. After the mice were sacrificed, tumor specimens were collected for further analysis.

## 3. Results

### 3.1. ASCs Isolated from Omentum Express Mesenchymal and Stem Markers

ASCs isolated from the omentum initially displayed a circular morphology but transitioned to a typical spindle-shaped morphology with attached fibroblasts after 90 h of culture, as shown in Figure 1A. To characterize the expression of cell surface markers, ASCs were examined using flow cytometry. ASCs expressed surface markers supporting their stemness and mesenchymal stem cell features, such as CD29, 44, 90, and 105, but did not express the hematopoietic cell-related antigens CD14, 34, or 45, nor did they express the epithelial cell-related antigen EpCAM (CD326) (Figure 1B).

### 3.2. ASCs Express CAF Markers and Promote Tumor Growth and Fibrosis of the Cancer Stroma

The expression of FAPα and αSMA, representative markers of CAFs, in ASCs was evaluated by immunofluorescence staining. The expression of FAPα and αSMA in the plasma membrane and cytoplasm were confirmed as shown in Figure 2A. Quantitative analysis of FAPα and αSMA expression in ASCs revealed that approximately 90% of cells were positive for these markers (αSMA: 91.3 ± 5.2%, FAPα: 88.9 ± 6.8%, n = 5 independent experiments). To address potential differences in proliferation rates between MKN45 cells and ASCs, we conducted a 72 h proliferation assay. The results showed no significant difference in proliferation rates between MKN45 cells and ASCs (passage 2) over this time period (Appendix A). Next, we evaluated the effect of ASCs on the fibrosis of the GC stroma in a mouse xenograft model co-inoculated with MKN45 cells. The time course of the subcutaneous tumor volume is shown in Figure 2B. Tumors formed from the inoculated MKN45 cells plus ASCs were significantly larger than those from MKN45 cells alone, with the difference becoming statistically significant 7 days after inoculation (*p* < 0.05). Mallory-Azan staining revealed that fibrosis was significantly larger in tumors created from MKN45 cells and ASCs than in the control group (*p* < 0.01; Figure 2C). In the MKN45 cells alone group, a minor fibrosis was observed, likely due to mouse-derived fibroblasts. In contrast, in the MKN45 cells co-inoculated with ASCs group, a more pronounced fibrosis was noted. Expression of αSMA was also higher in animals implanted with MKN45 plus ASCs, which corresponds to the notable increase in fibrous tissue. Importantly, ASCs labeled with the PKH26 cell linker were observed in the αSMA-stained regions, suggesting that human-derived ASCs contributed to the formation of the fibrotic area (Figure 2C). These results suggest that omentum-derived ASCs may differentiate into CAFs and contribute to the formation of a fibrotic tumor microenvironment, which could support the development of peritoneal metastases in GC.

### 3.3. The Interaction of ASCs with Gastric Cancer Cells Increases Il-6 Production Levels

We subsequently investigated the cell–cell interactions between ASCs and GC cells. First, the effect of the co-culture of ASCs and GC cells on cytokine production was examined using a multiplex cytokine/chemokine array (Figure 3A). The dots on the array film showed that the levels of IL-6 were undetectable in MKN45 cells alone. However, IL-6 secretion increased in the co-culture of ASCs and MKN45 cells. Of the 36 factors examined, a similar pattern was observed for CCL2 and IL-18. However, the increase in IL-6 was the most pronounced. To further quantify these observations, we examined the IL-6 levels in various culture conditions using ELISA (Figure 3B). GC cells alone (MKN45, MKN74, and NUGC4) secreted very low levels of IL-6. ASCs secreted higher levels of IL-6 than all three types of GC cells. Notably, when ASCs were cultured with CM from GC cell lines, IL-6 production increased significantly. This increase was not merely additive but synergistic, with IL-6 levels in the co-culture and ASCs cultured with GC CM being significantly higher than the sum of IL-6 levels from ASCs alone and GC cells alone (*p* < 0.01). This synergistic effect was observed for all three GC cell lines tested. These results suggest that the interaction between ASCs and GC cells stimulates a marked increase in IL-6 production, primarily from ASCs.

### 3.4. Effect of ASCs on Proliferation of Gastric Cancer Cells

We investigated the effect of ASCs on the proliferative ability of GC cells using an MTT assay in an indirect co-culture model (Figure 4). No effect on proliferative capacity was observed in MKN74 and NUGC4 cells when co-cultured with ASCs; however, cell proliferation was significantly enhanced in MKN45 cells (*p* < 0.05). The effect of ASCs on promoting GC cell proliferation was not as pronounced as expected.

### 3.5. ASCs Enhance Self-Renewal of Gastric Cancer Cells

We performed a sphere formation assay to investigate the effects of ASCs on the tumor-initiating ability of GC cells. Sphere-formation assays have been extensively utilized to retrospectively recognize CSCs based on their ability to evaluate self-renewal at the single-cell level in vitro [15]. Prior to assay, ASCs were stained with the red fluorescent dye PKH26. We found that GC cells cultured alone formed spheres in MKN45 and MKN74 cells but not in the NUGC4 cells. However, ASCs alone did not form a sphere. We found that the sphere-forming efficiency of MKN45 and MKN74 cells was significantly increased (*p* < 0.01) when directly co-cultured with ASCs, and representative images showed that PKH26-positive ASCs survived and were involved in GC spheres, as shown in Figure 5A.

Subsequently, we examined the expression of CD44v6, a key cell surface marker for cancer stem cells, and the self-renewal proteins SOX2, Nanog, and Oct4 in MKN45, MKN74, and NUGC4 cells after being co-cultured with ASCs or cultured alone. After co-culturing, GC cells were separated from ASCs before protein extraction and analysis. In GC cells co-cultured with ASCs, the protein levels of CD44v6 and Nanog increased, especially in MKN45 and MKN74 cells, whereas the levels of Oct4 and SOX2 remained stable in all cell lines (Figure 5B).

### 3.6. ASCs Increase Nanog and CD44v6 Expression in Gastric Cancer Cells Through IL-6 Secretion

To determine the role of IL-6 in mediating the acquisition of cancer stemness enhanced by ASCs, we investigated the effect of IL-6 neutralizing antibody on the sphere-forming capacity of GC cells. Consistent with previous results, the number of tumor spheres increased when co-cultured with ASCs. However, IL-6 neutralizing antibody (0.1 μg/mL) significantly reduced the sphere formation of GC cells by ASCs (Figure 6A). Western blotting was performed to determine the effects of IL-6 neutralizing antibody on CSC markers. The expression of Nanog and CD44v6 was relatively increased in MKN45 cells treated with human recombinant IL-6 (50 ng/mL) and co-cultured with ASCs. As expected, IL-6 neutralizing antibody markedly diminished the ASC-induced upregulation of Nanog and CD44v6 expression in MKN45 cells (Figure 6B). These results indicated that IL-6 is an important regulator that contributes to the increased tumor stemness of GC cells induced by ASCs.

### 3.7. ASCs Enhance Chemotherapy Resistance of Gastric Cancer Cells

Numerous studies have demonstrated that CSCs are resistant to chemotherapy [16]. Thus, we examined whether ASCs influence the sensitivity of GC cells to the widely prescribed chemotherapeutic agent 5-FU. MKN45 cells alone (MKN45 group) or MKN45 + ASCs (MKN45 plus ASCs group) were subcutaneously injected into the backs of mice to form xenografts. After tumors reached 100–150 mm^3^ in volume, the mice were randomized to receive treatment with PBS or 20 mg/kg 5-FU five times per week for two weeks. In the MKN45 group, control tumors treated with PBS grew to >500 mm^3^ in the 14 days after treatment and 5-FU significantly reduced the tumor volume to an average of 261 mm^3^ (48.1% less than control tumors, *p* < 0.05). In the MKN45 plus ASCs group, control tumors grew faster than those in the MKN45 group and reached over 800 mm^3^ in 14 days following treatment. The effect of 5-FU was not statistically significant (6.8% less of control tumors, *p* = 0.37), although it showed an anti-tumor trend (Figure 7A).

The xenografts were removed and analyzed after 5-FU treatment was terminated. Co-inoculation with MKN45 cells and ASCs significantly reduced apoptotic cells by 5-FU treatment compared with MKN45 cells alone, as measured by the TUNEL assay (*p* < 0.01, Figure 7B). Furthermore, the number of cells expressing both CD44v6 and Nanog was higher in the MKN45 + ASCs group than in the MKN45 group (Figure 7C). Immunohistochemical staining also showed higher levels of IL-6 in MKN45 + ASCs tumors than in MKN45 tumors (Figure 7D).

## 4. Discussion

In this study, we analyzed ASCs derived from the omentum, the primary intraperitoneal reservoir of adipose tissue, to investigate the effects of GC on PM formation. Evidence indicates that ASCs derived from subcutaneous adipose tissue exert tumor-promoting effects in various cancer types [17,18,19], whereas conflicting reports have revealed the anti-tumorigenic potential of ASCs, such as anti-proliferative and pro-apoptotic elements [20,21,22]. This discrepancy may be attributed to the ASC subtype. Adipose tissue is traditionally classified into subcutaneous and visceral omental adipose tissues, and growing research has focused on the differences in molecular functions and genotypes between subcutaneous ASCs and omental ASCs [23,24,25]. These differences suggest that omentum-derived ASCs may play distinct roles in the progression of intraperitoneal cancers. To our knowledge, this is the first study to investigate the effects of omental ASCs on GC progression.

While we observed larger tumors in the MKN45 + ASCs group compared to MKN45 alone, we acknowledge that potential differences in proliferation rates between these cell types could contribute to this effect. However, our 72 h in vitro proliferation assay showed no significant difference between MKN45 cells and ASCs. It is important to note that in vitro proliferation rates may not directly translate to in vivo tumor growth due to the complex tumor microenvironment. Our study primarily focuses on the qualitative changes in the tumor microenvironment induced by ASCs, particularly their role in promoting desmoplasia and cancer stemness, rather than their direct contribution to tumor size through proliferation.

We found that omentum-derived ASCs express key CAF markers, such as FAPα and αSMA, and contribute to desmoplasia in a model of co-inoculation with GC cells. PM in GC is characterized by an abundant fibrotic stroma with CAFs which impairs drug delivery, inhibits immune cell accessibility, and promotes tumor progression and treatment resistance [26]. These results have led us to speculate that intraperitoneal ASCs contribute to the resistance to PM treatment.

Adipose tissue secretes numerous cytokines, including IL-6, tumor necrosis factor (TNF)-α, IL-1β, IL-8, and monocyte chemoattractant protein (MCP)-1, which are primarily produced by non-adipocyte cells within the adipose depot [27,28]. We found that the interaction between omentum-derived ASCs and GC cells stimulated the secretion of IL-6 from the ASC, which is a proinflammatory cytokine that features a wide range of bioactivities during tumorigenesis and metastasis [29,30]. Ham et al. reported that IL-6 in GC tissues was mainly co-expressed with stromal-related genes and showed IL-6 expression in stromal cells but not in cancer cells in human GC tissues [31]. Based on these findings, our results suggest that ASCs may be the main source of IL-6 in PM lesions.

CSCs play crucial roles in both chemoresistance and recurrence owing to their high DNA repair capacity, apoptosis resistance, and self-renewal potential. In addition to their specific intrinsic signaling pathways, CSCs also acquire extracellular cues from the TME [32,33]. We demonstrated that GC cells co-cultured with ASCs enhanced their ability to form spheres with CSCs properties. We also found that GC cells co-cultured with ASCs enhanced the expression of Nanog and CD44v6 compared to the GC cells cultured alone. The homeobox domain transcription factor Nanog, a key regulator of embryonic development and cellular reprogramming, has been reported to be broadly expressed in human cancers, including GC [34]. Numerous studies have reported that Nanog promotes cancer stemness via the IL-6/signal transducer and activator of transcription (STAT)3 signaling and is an accurate indicator of poor prognosis in cancer patients [35].

CD44, a transmembrane receptor for hyaluronic acid, has also been used as a cell-surface marker for CSCs [36]. Among the CD44 isoforms, CD44v6 has been shown to play a major role in cancer progression due in part to its ability to directly bind to major cytokines produced in the TME [37,38]. CD44 is anticipated to be the most promising biomarker for the diagnosis and treatment of GC [39,40]. Nanog contributes to the maintenance of the stem cell-like properties of CD44v6-expressing cancer cells in hepatocellular carcinoma [41]. Our data suggest that ASCs-derived IL-6 enhanced Nanog and CD44v6 expression and maintained GC cell stemness.

The mechanisms by which ASCs contribute to the stemness and chemoresistance of GC cells are likely multifaceted. IL-6 secreted by ASCs may activate the JAK/STAT3 pathway in GC cells, leading to increased expression of stemness-related genes such as Nanog and CD44v6 [42]. Additionally, ASCs may secrete other factors, such as growth factors or exosomes, that further promote cancer stemness and drug resistance [43]. The interaction between ASCs and GC cells may also lead to changes in the extracellular matrix composition, creating a niche that supports cancer stem cell maintenance and protects them from chemotherapy-induced apoptosis [44].

Chemotherapy resistance is the most significant obstacle to PM treatment. In our study, a xenograft mouse model co-inoculated with MKN45 cells and ASCs showed enhanced expression of CD44v6 and Nanog, along with markedly reduced apoptosis in response to 5-FU treatment compared to a model inoculated with MKN45 cells alone. This observation suggests that ASCs not only promote cancer stemness but also directly contribute to chemoresistance. The increased expression of CD44v6 and Nanog may enhance DNA repair mechanisms and anti-apoptotic signaling in cancer cells, making them more resistant to 5-FU-induced cell death [45]. Furthermore, ASCs may alter the tumor microenvironment in ways that reduce drug penetration or activate alternative survival pathways in cancer cells [46].

We investigated the effect of 5-FU in the present study because 5-FU combination regimens have been prescribed worldwide as the first-line chemotherapy regimens for unresectable GC patients [47]. Although omental ASCs have previously been reported to inhibit the apoptotic effects of cisplatin and paclitaxel in ovarian cancer cells [48,49], our study is the first to demonstrate that omentum-derived ASCs induce 5-FU resistance in PM in GC. Future studies should focus on elucidating the specific molecular pathways involved in ASC-mediated chemoresistance and exploring potential strategies to overcome this resistance, such as targeting IL-6 signaling or other key mediators of the ASC–cancer cell interaction.

This study has several limitations. First, there is a lack of direct in vivo evidence regarding the causal relationship between ASC-secreted IL-6, the upregulation of Nanog and CD44v6, and 5-FU resistance. While our in vitro findings strongly support this mechanism, the complex tumor microenvironment may involve additional factors contributing to chemoresistance. Future studies should include in vivo experiments with IL-6 neutralization, comparing the effects of 5-FU treatment on tumors co-inoculated with MKN45 cells and ASCs to further clarify the role of IL-6 signaling in cancer stemness and resistance. Secondly, this study did not compare the biological properties of ASCs derived from sub cutaneous adipose tissue with those from the omentum. Although systemic mobilization of ASCs from adipose tissue has been suggested in mouse models [50,51,52], adipose tissue adjacent to the tumor, such as the omentum, is likely to be a more significant contributor to ASC presence [53]. Therefore, we focused specifically on interactions between omentum-derived ASCs and gastric cancer in this study, leaving the comparison to other adipose sources for future research. Lastly, while our study utilized the widely recognized CAF markers αSMA and FAP to demonstrate the differentiation of ASCs into CAFs, we acknowledge that CAFs represent a heterogeneous population. A more comprehensive characterization using additional markers such as Pdgfra, Myh11, Pecam, and CD45 could provide deeper insights into the CAF subtypes derived from ASCs within the tumor microenvironment, which should be considered in future investigations.

## 5. Conclusions

Our findings suggest that ASCs differentiate into CAFs within the cancer stroma of PM and that IL-6 secreted by ASCs induces 5-FU resistance through the upregulation of Nanog and CD44v6 in the PM of GC. This study enhances our understanding of the role of ASCs in PM treatment resistance and highlights the potential for new treatment strategies targeting ASCs. Additionally, future research will focus on elucidating the impact of ASCs on the formation of PM in GC.

## Figures and Tables

**Figure 1 cancers-16-04275-f001:**
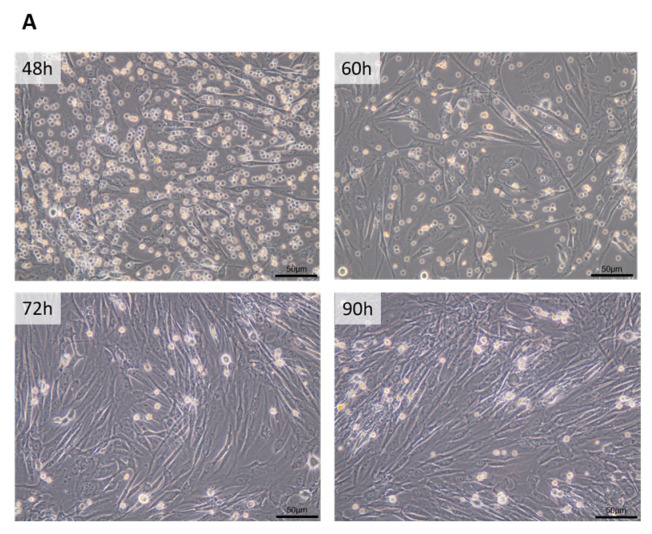
(**A**) A representative image of morphological changes in ASCs isolated from omental adipose tissues was visualized by phase contrast microscopy at a magnification of ×200. (**B**) Flow cytometry characterization of surface marker expression in ASCs. The cells are positive for CD2, CD44, CD90, and CD105 and negative for CD14, CD34, CD45, and CD326.

**Figure 2 cancers-16-04275-f002:**
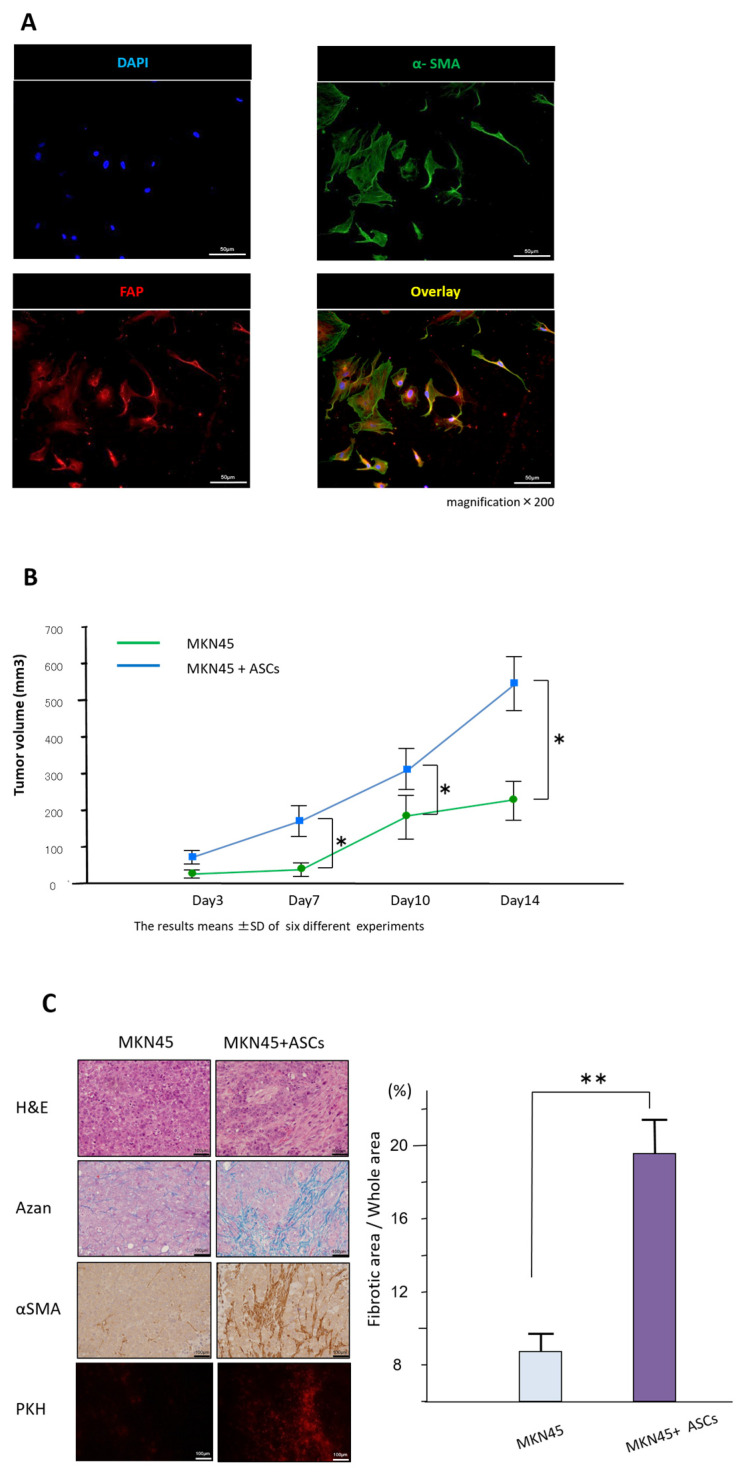
(**A**) Representative photomicrographs of immunofluorescence staining for fibroblast activation protein α (FAPα) and α-smooth muscle actin (αSMA). Original magnification ×200. (**B**) Comparison of mean of subcutaneous tumor volume between MKN45 cells co-inoculated with ASCs and those inoculated with MKN45 cells alone using the mouse xenograft model over 14 days after inoculation (n = 6 per group). (* *p* < 0.05). (**C**) Microscopic findings of mouse xenograft tumors. Histological examination was performed by hematoxylin and eosin (H&E) staining. Fibrotic tissue was determined using Mallory-Azan staining in the subcutaneous xenograft tumors. αSMA expression was evaluated by immunohistochemistry. The presence of ASCs in subcutaneous xenograft tumors was observed by fluorescent staining of PKH26 (red) (original magnifications, ×100). The fibrous area was measured semi-quantitatively. The fibrosis rate was significantly higher in the MKN45 cell and ASCs co-inoculation group compared to the control (MKN45 alone) group (** *p* < 0.01). The results means ± SD of five different experiments.

**Figure 3 cancers-16-04275-f003:**
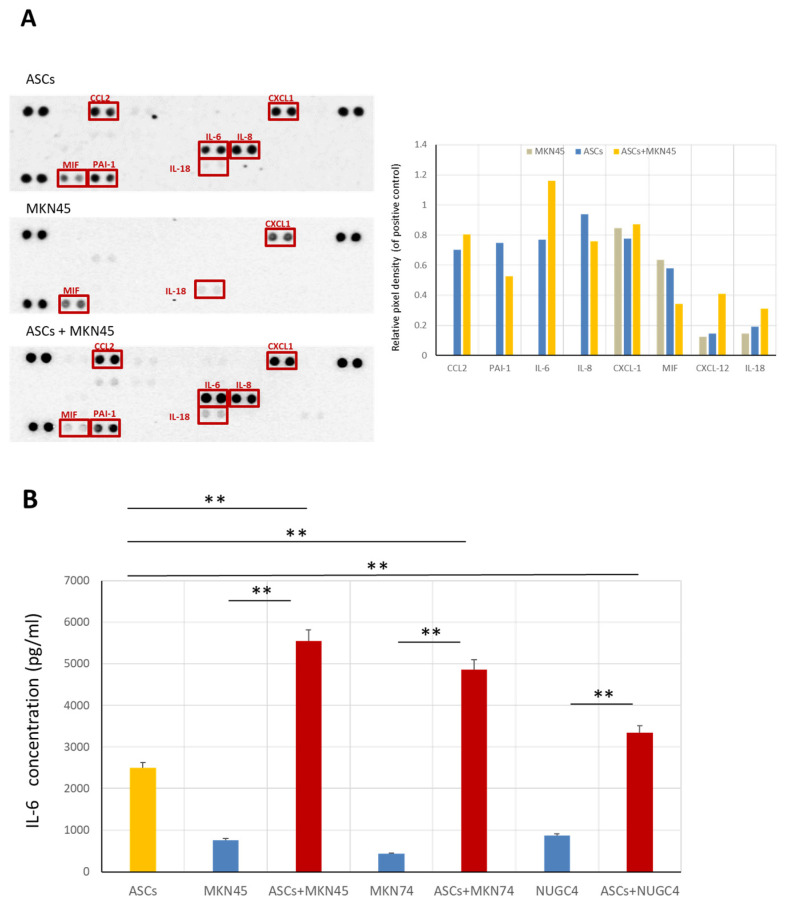
(**A**) Human cytokine array detecting multiple cytokines, chemokines, growth factors in a cell culture medium. ASCs were cultured alone or co-cultured indirectly with MKN45 cells in serum-free media for 48 h. The levels of various factors in culture media were measured using a proteome profiler array and normalized to positive control according to the manufacturer’s protocol. Data are expressed as fold inductions from duplicate experiments. (**B**) IL-6 concentrations secreted from ASCs were examined using ELISA. A conditioned medium from all GC cell lines significantly promoted IL-6 secretion from ASCs. Data represent the mean ± SD of triplicate wells for three independent experiments (** *p* < 0.01).

**Figure 4 cancers-16-04275-f004:**
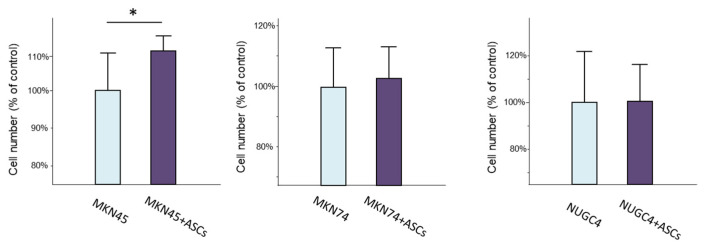
The proliferative capacity of GC cells (MKN45, MKN74, NUGC4) co-cultured with ASCs was evaluated using MTT assay. Cell proliferation was compared to GC cells cultured alone. The experiment was conducted three times, with each sample being assessed in triplicate. Data are expressed as the means ± SD. (* *p* < 0.05).

**Figure 5 cancers-16-04275-f005:**
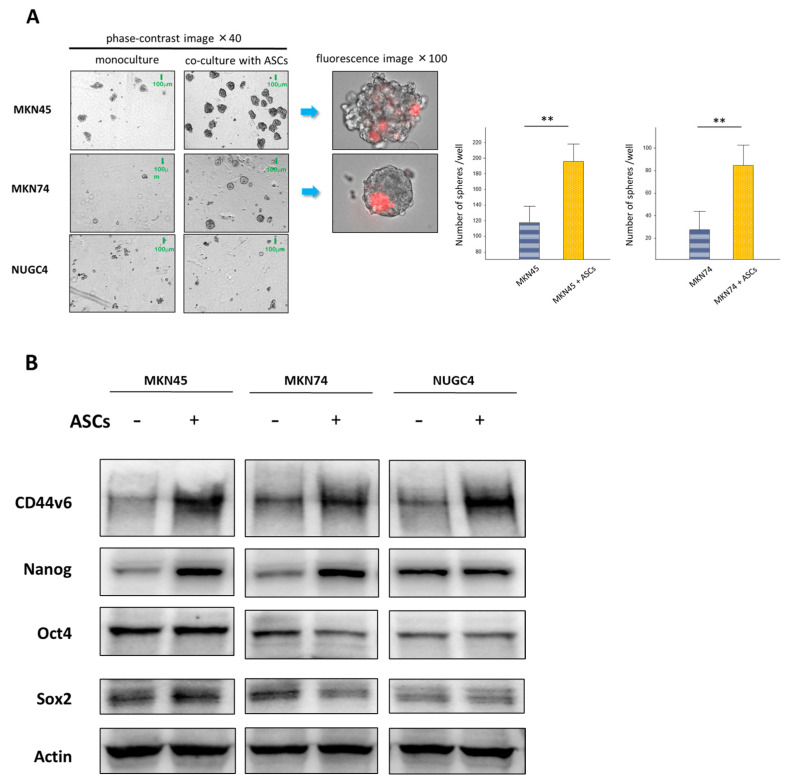
(**A**) Representative images of sphere formation in GC cells and PKH26 immunofluorescence staining in ASCs. Direct co-culturing with ASCs significantly promoted the sphere formation ability of MKN45 and MKN74 cells (** *p* < 0.01). Only spheres larger than 50 μm in size were included in the analyses. (**B**) A Western blotting analysis evaluating the protein levels of CD44v6, Nanog, Oct4, and SOX2 in GC cells after being co-cultured with ASCs compared to GC cells that were cultured alone. GC cells were separated from ASCs after co-culturing before protein extraction. β-actin was used as a normalized control. The uncropped blots are shown in Appendix A.

**Figure 6 cancers-16-04275-f006:**
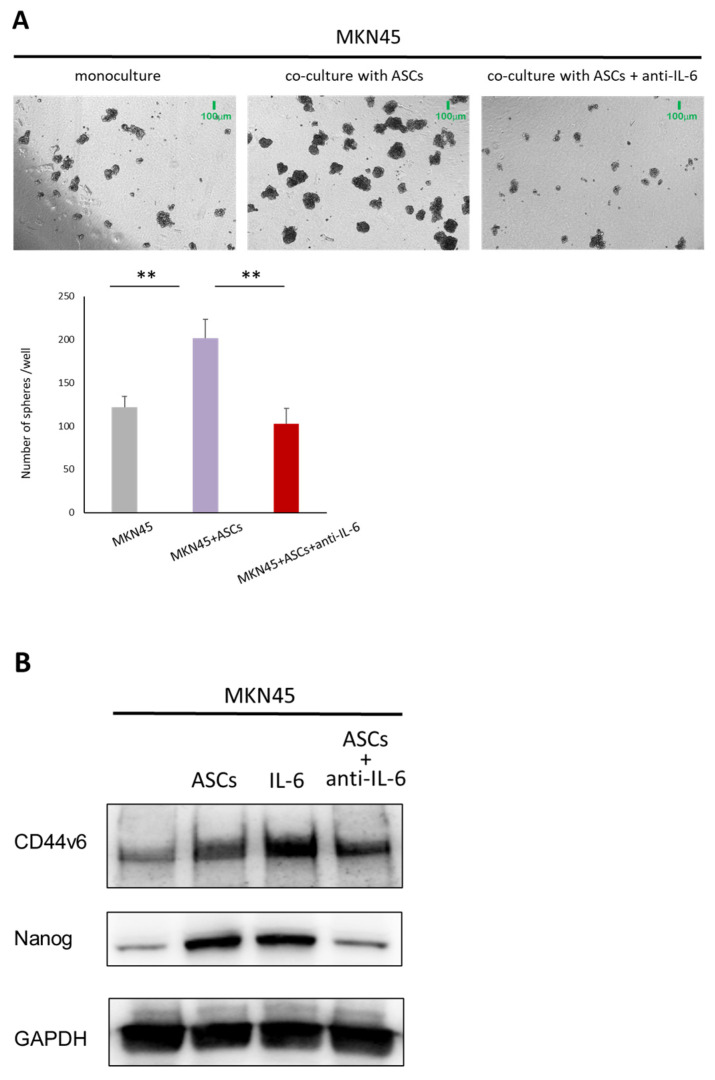
(**A**) Representative images and the quantification of spheres generated by MKN45 cells co-cultured with ASCs and MKN45 cells cultured alone. A neutralizing antibody against IL-6 (anti-IL-6, 0.1 μg/mL) was added to co-cultures of ASCs and MKN45 cells. Results are expressed as the mean ± SD (** *p* < 0.01). (**B**) A Western blotting analysis showing the expression of the indicated proteins in GC cells co-cultured with ASCs and treated with human recombinant IL-6 (50 ng/mL) and IL-6 neutralizing antibody (0.1 μg/mL) for 72 h. GAPDH was used as a normalized control. The uncropped blots are shown in Appendix A.

**Figure 7 cancers-16-04275-f007:**
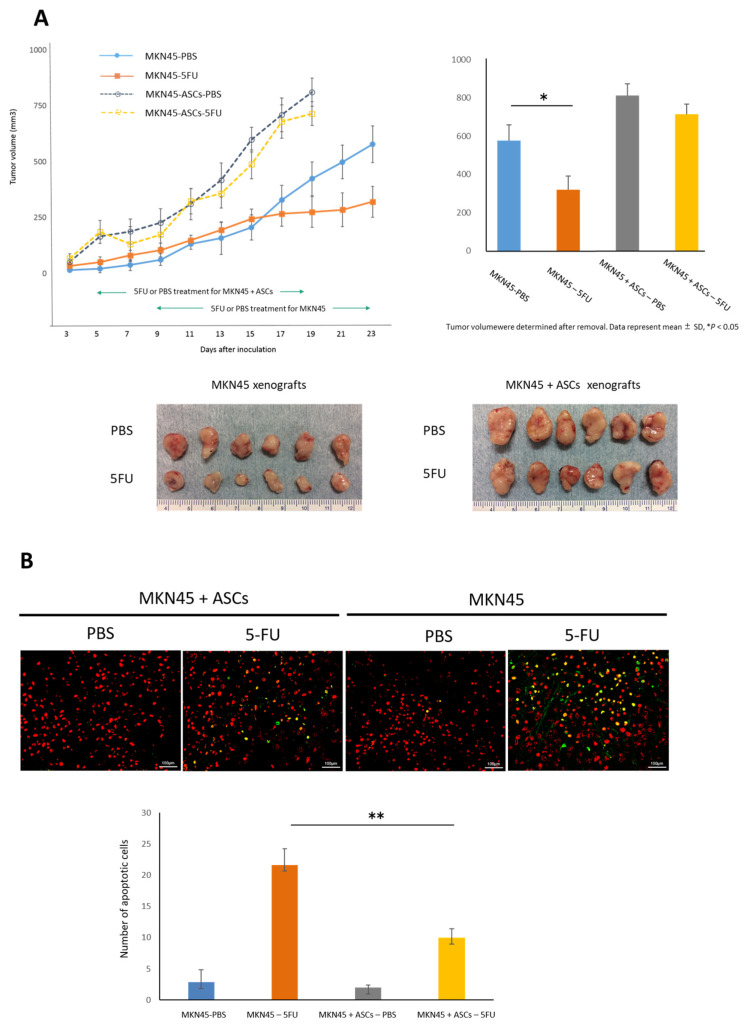
(**A**) Subcutaneous tumor growth curves for MKN45 alone and MKN45 plus ASCs xenografts treated with PBS or 5-FU. Results are expressed as the mean ± SD (* *p* < 0.05). The lower panels showed xenograft tumors removed from the respective treatment groups. (**B**) Representative images of the TUNEL assay for apoptosis (upper panels; original magnification, ×100). The number of apoptotic cells in xenograft tumors was measured and is shown as the average count from five non-overlapping tumor areas (lower panel). Results are expressed as the mean ± SD (** *p* < 0.01). (**C**) Microscopic images following the immunofluorescent analysis of tumors using CD44v6 (green) and Nanog (red) (original magnification, ×100). (**D**) Light microscopic images of tumor tissue sections following immunohistochemical staining of IL-6 (original magnification, ×100).

## Data Availability

The original contributions presented in the study are included in the article; further inquiries can be directed to the corresponding author.

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
