# Peer review of "Crosstalk Between Omental Adipose-Derived Stem Cells and Gastric Cancer Cells Regulates Cancer Stemness and Chemotherapy Resistance"

_cancers, 2024, doi:10.3390/cancers16244275_

Round 1

Reviewer 1 Report

Comments and Suggestions for Authors

Line 202: [?]

Line; 206: [OK?]

Line 252-260: Are the MKN45 cells and ASCs proliferate rate similar or different?

The different cell proliferation rates may affect the tumor size, although the total injection cell numbers of MKN45 cells alone and MKN45 cells with ASCs cells are the same.

Line 260: The conclusion is lacking evidence.

Line 261: Figure 2B: tumor images.

Line 261: Figure 2C: FAP staining? Or Ku70 staining since a-SMA can react both with humans and mice. Do the cells consist of the fibrotic area of the MNK45 alone group mouse original? Does comparing the fibrotic area of MNK45 alone (mouse original) and MKN45 cells with ASCs cells (human original) represent scientific meaning?

Line 278: Multiplex cytokine/chemokine array? Source and catalog number.

Line 286: Figures 3A and 3B,  x and y axes should have marks to indicate the represented numbers.

Line 292: The description is no evidence. Could you provide evidence that the IL6 levels increase from ASCs?

Line 324: Figure 5B needs to include ASCs alone to evaluate whether the protein expression increased due to ASCs or not since the protein extraction was not GC alone.

Line 410: [OK?]?

Line 423 and Line 425: [Reference(s)]?

Line 452-453: [IL-6 secreted by ASCs induces 5-FU resistance through the upregulation of Nanog and CD44v6 in the PM of GC]? No evidence demonstrate that IL6 induces GC increasing Nanog and CD44, causing 5-FU resistance.  To verify that in vivo study need includes two groups  (MKN45+ASCs+anti-IL6) treated with PBS and 5-FU and evaluate the tumor growth status.

Comments on the Quality of English Language

Minor editing of English language required.

Reviewer 2 Report

Comments and Suggestions for Authors

The manuscript explained the interaction and related cell signaling between adipose-derived stem cell and gastric cancer cell, which is a good angle to understand the cellular processing of gastric tumor cell growth in metastasis, particularly in peritoneal adipose-enriched omentum. Moreover, authors showed the chemotherapy resistance by ASCs contribution. However, the study lacks some crucial assay and need some improvements. Here are my concerns which should be addressed as below:

For the stemness and mesenchymal features of ASCs, authors should check proliferating markers and adipose cell markers other than CD29, CD44 and CD90. They are critical evidences for ASCs.

In fig. 2A, authors showed the immunofluorescence staining for the CAF markers. However, these markers are not representative and limited for the complicated CAF populations which may contain immune cells, endothelial cells, myocytes and pericytes. Authors should check more markers to clarify the CAF population, such as Pdfgra, Myh11, Pecam and CD45. Additionally, how many percentage of ASCs express FAPα and αSMA? In your fig. 2A, all cells are FAPα and αSMA positive, but αSMA as a classic smooth muscle cell marker has rare or limited expression in adipose cell.

In Fig. 3A, authors showed a minor increase of IL-6 in MKN45+ASCs group by cytokine array. However, the ELISA for IL-6 in Fig. 3B exhibited more than 2 fold change in MKN45+ASCs. Also, no IL-6 detected in cytokine array for MKN45 group but a partial concentration of IL-6 in ELISA. Authors should clarify the discrepancy for these data.

In Fig. 5A, the brightness images showed the enhanced sphere-forming efficiency by co-culturing with ASCs, which is obvious and important. The sizes of sphere are also increased by images, while authors verified a minor effect on gastric cancer cell proliferation. I suggest that some proliferating markers should be performed rather than the simply quantitative results for cell number or size.

Authors have measured the proliferation and self-renewal of gastric cancer cells. How about the metastasis of MKN45 cells in the ASCs co-culture?

Comments on the Quality of English Language

Minor editing of English language required.

Reviewer 3 Report

Comments and Suggestions for Authors

The paper examines the role of omental adipose-derived stem cells (ASCs) in promoting gastric cancer stemness and chemotherapy resistance. It highlights the ASCs' potential to act as progenitors of cancer-associated fibroblasts, influencing the cancer stem cell niche and reducing apoptosis in tumor cells during chemotherapy, particularly with 5-FU.

1. It would be beneficial to add control experiments using ASCs treated with an IL-6 inhibitor to isolate the effect of IL-6 signaling on cancer cell behavior.

2. The author reports increased IL-6 production and changes in CSC marker expression. Is there an in vitro quantitative analysis of how these changes correlate with changes in tumor size and chemoresistance. 

3. More discussion is needed to explain the mechanisms by which ASCs contribute to the stemness and chemoresistance of gastric cancer cells.

4. line 410: Please delete the [OK?]

Round 2

Reviewer 1 Report

Comments and Suggestions for Authors

Thanks for the authors’ response. The suggestions for the manuscript are listed below.

1.       Please include the result of a 72-hour proliferation assay of MKN45 cells and ASCs in Figure 2 or supplementary Figure 1.

2.       Please provide the scale bars: Figure 1A, 2A, 2C, 7B and 7C.

3.       The main text or figure legend of Figure 5B should point out that western blotting results were from GC cells, not co-cultured cells.

Reviewer 2 Report

Comments and Suggestions for Authors

The authors addressed all my concerns properly, and I have no more questions for the study. The revised manuscript is much better and it can be published.

Author Response

Thank you for your positive feedback and thorough review of our manuscript. We greatly appreciate the time and effort you have invested in evaluating our work.

Your constructive comments and suggestions have been invaluable in improving the quality and clarity of our manuscript.We are pleased to hear that our revisions have adequately addressed all of your concerns and that you find the revised manuscript suitable for publication. Your approval is very encouraging, and we are grateful for your support.